# Spray-Drying Encapsulation of the Live Biotherapeutic Candidate *Akkermansia muciniphila* DSM 22959 to Survive Aerobic Storage

**DOI:** 10.3390/ph15050628

**Published:** 2022-05-20

**Authors:** Joana Cristina Barbosa, Diana Almeida, Daniela Machado, Sérgio Sousa, Ana Cristina Freitas, José Carlos Andrade, Ana Maria Gomes

**Affiliations:** 1Universidade Católica Portuguesa, CBQF—Centro de Biotecnologia e Química Fina—Laboratório Associado, Escola Superior de Biotecnologia, Rua Diogo Botelho 1327, 4169-005 Porto, Portugal; jcbarbosa@ucp.pt (J.C.B.); dialmeida@ucp.pt (D.A.); sdcsousa2@gmail.com (S.S.); amgomes@ucp.pt (A.M.G.); 2TOXRUN—Toxicology Research Unit, University Institute of Health Sciences, CESPU, CRL, 4585-116 Gandra, Portugal; jose.andrade@iucs.cespu.pt

**Keywords:** aerobic storage, *Akkermansia muciniphila*, microencapsulation, simulated gastrointestinal passage, spray-drying, stability

## Abstract

*Akkermansia muciniphila* is regarded as a promising next-generation probiotic or live biotherapeutic candidate. Effective delivery strategies must be developed to ensure high enough viability of the probiotic strain throughout its industrial formulation, distribution chain, shelf-life, and, ultimately, the host’s gastrointestinal tract, where it should exert its beneficial effect(s). Among the possible methodologies, spray-drying is considered industrially attractive regarding its costs, efficiency, and scalability, with the due parameter customization. In this study, spray-drying was explored as a one-step process to encapsulate *A. muciniphila* DSM 22959, testing the drying settings and three different dairy-based matrices. Microcapsule morphology and size was assessed, and viability throughout storage at 4 or 22 °C and simulated gastrointestinal passage was determined. *Akkermansia muciniphila* microencapsulation by spray-drying, using 10% skim milk and inlet/outlet temperatures of 150/65 °C, is effective in terms of viability stabilization, both during prolonged aerobic storage and exposure to simulated gastrointestinal passage. *Akkermansia muciniphila* viability was maintained at around 10^7^ CFU/g up to 28 days at 4 °C under aerobic conditions with viability losses inferior to 1 log reduction. This methodology provides the necessary conditions to efficiently deliver the recommended dose of live *A. muciniphila* in the human gut as a live biotherapeutic product.

## 1. Introduction

The knowledge regarding the pivotal role of intestinal commensal bacteria in human health has been increasing in the past few years, with a growing interest of the scientific community in exploiting the beneficial properties of such species with potential application as probiotics or as live biotherapeutics [1,2]. Among those, *Akkermansia muciniphila* is one of the most promising candidates proposed as a next-generation probiotic. This commensal bacterium has a high potential to be incorporated into foods or pharmaceutical formulations due to its demonstrated relevant biological effects in several metabolic conditions [3,4].

*Akkermansia muciniphila* is an anaerobic, Gram-negative, and mucin-degrading bacterium [5], which represents 1 to 3% of human fecal microbiota in healthy adults [6]. Recent studies show a positive correlation between the relative abundance of *A. muciniphila* in the gut and a general healthy state [7,8,9]. For instance, the potential beneficial effects of *A. muciniphila* supplementation in several metabolic parameters of overweight/obese insulin-resistant human volunteers, namely cardiovascular risk factors, were recently demonstrated in a safe and well-tolerated intervention [10].

While promising, the intended positive health effects of *A. muciniphila* must be guaranteed, and as in most probiotic candidates, that begins by securing the survival of an adequate numbers of viable cells—recommended to be 10^9^ colony forming units (CFU) per dose—throughout storage and until the end of the gastrointestinal tract (GIT) [11]. In such a way, the incorporation into carrier formulations emerges as a key strategy to confer them protection (sensitive to oxygen) during the usually aerobic storage [3,12].

Encapsulation is gaining interest among the industry and the scientific community as a strategy to preserve bacterial cell viability, as it involves the cellular entrapment/coating within a material or mixture of materials [13]. Among the several methodologies that can be used to encapsulate probiotic bacteria, drying techniques are usually preferred as this process decreases the formulations’ water content, and, more specifically, spray-drying is one of the most popular due to appealing characteristics in terms of operation, scale-up, costs, and efficiency [14]. Several operational factors must be carefully considered and improved, depending on the nature of the probiotic bacterium itself and the intended final product and which involve the selection of the ideal protective matrix and the adjustment of the drying parameters [15]. Briefly, the spray-drying process consists of the atomization of bacterial suspensions into droplets due to the action of a controlled flow of hot air, producing dry spherical powder particles and enabling the dehydration of large amounts of liquid feed cultures in a short period of time [16]. One of the major disadvantages of the application of this methodology to commensal bacteria is the continuous exposure to oxygen during the process [14,17]. Regarding *A. muciniphila*, this species was originally classified as a strict anaerobe [5], but it was recently demonstrated that it can tolerate small amounts of oxygen, being reclassified as an aerotolerant anaerobe [18,19]. This makes *A. muciniphila* a good candidate for the application of spray-drying methodologies, which will facilitate its future application as a live biotherapeutic [20]. Indeed, a previous work by Chang and colleagues explored the application of spray-drying to this species, with promising results for a potential industrial application. However, in this work, a synthetic matrix was employed (chemically modified alginate); moreover, only anaerobic, refrigerated storage conditions were assessed [20]. Other factors that pose a viability loss risk during the drying process are the high temperature conditions and dehydration [13]. As such, the present work aims to: (i) establish a suitable procedure for *A. muciniphila* DSM 22,959 encapsulation using the spray-drying technique, without prior upstream processes, by maximizing the drying settings (inlet and outlet air temperatures variation) in combination with easily available protective matrices; (ii) evaluate spray-dried *A. muciniphila* viability during storage under aerobic conditions at different temperatures, with identification and characterization of the most effective product; and (iii) assess the performance of the selected product under simulated gastrointestinal tract (GIT) conditions.

## 2. Results and Discussion

Three different matrices were selected for *A. muciniphila* spray-drying: skim milk (SM), whey protein concentrate (80%; WPC), and whey protein isolate (90%; WPI) based on easy access and customary use in spray-drying of conventional probiotics [21,22]. Besides the matrix effect, the inlet temperature and outlet temperature effects on the viability of spray-dried *A. muciniphila* DSM 22,959 bacterial cells were also evaluated.

### 2.1. Spray-Drying Process Yields and Cell Viability

The initial matrix preparations possessed around 10% of total solids content, considering the matrix and the bacterial biomass. The percentage of recovered dried powder for each condition is presented in Table 1 and seems to be independent from the matrix itself and the drying conditions, with a recovery rate always higher than 50% (SM #4) and up to 80% (WPI #2) of the initial total solids content. Almost all yields fit the range defined by Broeckx and co-workers, who proposed a minimum yield of around 70% (m/m) to consider the process economically feasible [21].

Table 1 shows the total number of *A. muciniphila* cells that were initially incorporated into each matrix—ranging from 10^10^ and 10^11^—and the number of encapsulated cells at the end of the process, as well as the number of viable *A. muciniphila* cells per gram obtained for each processing condition. The highest number of encapsulated cells per gram was obtained for the skim milk matrix, with a reduction of approximately two log cycles for every processing condition. In particular, processing conditions #2 (150 and 65 °C inlet and outlet temperatures, respectively) and #3 (170 and 65 °C inlet and outlet temperatures, respectively) provide a significant protective effect among the tested conditions (*p* < 0.05). Indeed, although this condition did not yield the highest amount of dried powder, it is possible to observe that the skim milk matrix provided a significantly higher protection to *A. muciniphila* cells during the encapsulation process in comparison to the other two matrices tested (*p* < 0.05). Additionally, conditions #2 and #3—corresponding to the application of the lowest outlet temperature, 65 °C—consistently resulted in a higher *A. muciniphila* viability immediately after processing (Table 1), although this tendency does not seem to be statistically representative except for the already mentioned skim milk matrix. Moreover, this result seems not to be affected by the variation of the inlet temperature. Broeckx and colleagues (2020) evaluated the effect of various spray-drying settings—including inlet and outlet temperatures—on the viability of the probiotic bacterium *Lacticaseibacillus rhamnosus* GG. The authors reported a decrease on bacterial cells viability resulting from higher inlet and outlet temperatures, which is consistent with heat damaged cells during the drying process. Although the authors did not attempt to maintain the outlet temperature while varying the inlet, an in-depth analysis of the data shows a similar tendency, with close outlet temperatures showing comparable bacterial counts, despite the inlet temperature [21]. Similar results were also reported by Behboudi-Jobbehdar et al. and Ghandi et al. for the optimization of spray-drying conditions for probiotic strains from *Lactobacillus acidophilus* and *Lactococcus lactis* species, respectively [22,23].

Previous studies have shown the influence of inlet and outlet temperatures on the viability of probiotics strains (see [15] for a review). In particular, the outlet temperature was found to critically influence the post-drying viability of classical probiotics, with higher temperatures inducing higher viability losses. In the opposite direction, if the outlet temperature is too low, the resulting powders will retain higher moisture content and water activity, which will, in turn, reduce the expected shelf-life of the formulation, as it will be explained in the following section. Yet, case-by-case optimization is expected to be required depending on the bacterial strains, drying parameters, and matrices and intended applications [15].

### 2.2. Water Activity of Spray-Dried Microcapsules

The water activity (a_w_) is a parameter that measures the free, unbound water within a powder, which can be available for bacterial metabolism. This parameter is of utmost importance when considering long-term storage; reducing the a_w_ below the minimum levels required to sustain active growth (around 0.6–0.8) reduces bacterial metabolism, thus avoiding proliferation and preserving the matrix [21]. Yet, the water content should still be enough to retrieve the probiotic activity upon rehydration [17,24]. An a_w_ value of around 0.1–0.2 has been defined as ideal for long storage of probiotic products [15,21,25].

In our study, a_w_ was determined immediately after processing for all the matrices and processing conditions and was found to be between 0.23 and 0.33 in all cases (Table 2). The powders were then stored appropriately, with a controlled humidity of around 33–34% using a saturated solution of magnesium chloride. Considering that the relative humidity varies with temperature, this storage condition was chosen since it could be maintained in both storage temperatures tested [26]. Additionally, when a product is stored under a constant humidity, there is a tendency to reach an equilibrium, considering its a_w_ [27]. Thus, in this study, the surrounding storage environment was intended to keep the a_w_ constant, as much as possible, throughout time. Indeed, after the storage period, the a_w_ presented only slight variations, with more evident shifts towards the expected environmental relative humidity in WPC and WPI conditions (Table 2). These values are higher than the proposed for the ideal long-time storage conditions (around 0.1); however, these variations seem not to be directly correlated with the observed viability losses, which will be discussed in the following section.

### 2.3. Viability of Spray-Dried A. Muciniphila upon Aerobic Storage

A major consideration when envisaging the application of next-generation probiotics for human consumption is the need to guarantee the protection of the probiotic viability and the formulation stability, throughout the industrial manufacturing processes and delivery chain of the final product until it reaches the consumer. Indeed, control of parameters such as atmospheric oxygen levels and temperature to maintain stringent conditions poses not only a logistic challenge, but also high operating costs. Thus, the viability of spray-dried *A. muciniphila* stored under aerobic conditions, either refrigerated or around room temperature—4 and 22 °C, respectively—was assessed over a period of 28 days. As observed in Figure 1 (and Appendix A), skim milk shows a significantly higher protective effect over time at both temperatures (*p* < 0.05), but particularly at 4 °C (Figure 1a). Both WPC and WPI failed to maintain *A. muciniphila* viability at 22 °C independently of the operating conditions tested (Figure 1b,c) with WPC showing a loss of viability even at 4 °C for conditions #3 and #4, corresponding to the higher inlet temperature (170 °C). Preservation at lower temperatures was shown to be more beneficial for long-term storage of probiotic strains, such as *Bifidobacterium longum*, probably because it reduces the bacterial metabolism, thus preventing the matrix spoilage and the senescence of the bacteria themselves [28]. Regarding the highest protective effect of skim milk comparatively to the whey protein derivates, this might be due to the so-called “water replacement hypothesis”, which suggests that low molecular sugars, such as lactose—present in higher amounts in SM composition, when compared with that of WPC or WPI—have greater affinity with polar moieties of the material cells than water when the moisture content of the cellular environment is rather low. Thus, during the spray-drying process, while the water molecules evaporate, lactose molecules can occupy the vacant places, preserving cellular integrity and preventing protein aggregation and unfolding [29]. Indeed, lactose itself demonstrated to support the survival of *Lactococcus lactis*, despite the fact that this effect could be significantly enhanced by mixing it with other protectants [22]. However, studies demonstrate that the interaction between encapsulated probiotic strains and their encapsulating matrices is often strain-dependent and should be analyzed case-by-case [3,30]. Considering the drying conditions, for all matrices, but particularly for skim milk, condition #2—corresponding to 150 and 65 °C inlet and outlet temperatures, respectively—seems to perform consistently and significantly better in terms of viability maintenance throughout the assessed storage period (Figure 1; *p* < 0.05).

Previous works with *A. muciniphila* have also looked at storage stability. Van der Ark and colleagues reported a high viability loss on double emulsion microencapsulated *A. muciniphila* after only 72 h of anaerobic, refrigerated storage [31]. Another study performed by Chang and colleagues also applied the spray-drying technology to *A. muciniphila*, using complex synthetic matrices based on sodium alginate; the authors observed a protective effect of the matrices on the bacterial viability upon dissolution of the powdered cells, when comparing with the free cells but only during a storage period of only 12 days and under anaerobic, refrigerated conditions [20]. Marcial-Coba and co-workers combined extrusion microencapsulation and freeze-drying using different cryoprotectants (a mixture of sucrose and trehalose or agave sirup) and achieved a high viability (10^7^–10^8^ CFU/g) after 30 days in both refrigerated anaerobic and aerobic storage [32]. Thus, when comparing with these works, the main strength of the present study is that *A. muciniphila* viability was maintained at levels rounding 10^7^ CFU/g after 28 days of refrigerated storage under aerobic conditions but using a much better cost-efficient approach since it uses protecting matrices that are easily accessible and relatively cheap and a well-established technique. This pioneering study constitutes a promising storage approach that mimics the more conventional storage methods, under aerobic and refrigerated conditions [33].

### 2.4. Selection of the Best Spray-Drying Conditions and Characterization of A. Muciniphila Microcapsules

As mentioned, skim milk was the best protective matrix for spray-drying under the conditions of this study. Additionally, operating conditions #2 (inlet/outlet: 150/65 °C) and #3 (inlet/outlet: 170/65 °C) were similar in terms of stability. However, condition #2 yielded a higher protective effect during processing, probably since both inlet and outlet temperatures are milder, although the mass of the recovered powder was slightly lower. Thus, the microcapsules resulting from this processing condition were analyzed using SEM, showing a spherical-like, deflated, wrinkled appearance with no apparent fissures on the surface (Figure 2). This structure seems to be common to many powders produced by spray-drying and was proposed to be dependent of several variables, including the matrices and the drying temperatures [23,34]. The shriveled surface is thought to be a result of the rapid evaporation of the water during the drying process [34]. Most microcapsules presented a size that ranged between 2 and 10 µm, with the average being around 5.8 µm. The relative size distribution is shown in Figure 3.

All things considered, a new, scaled-up (500 mL) batch using 10% skim milk as a protective matrix was prepared and processed as in operating condition #2, namely, inlet temperature 150 °C, outlet temperature 65 °C, pump speed 35%, and a flow rate of 10.5 mL/min, corresponding to processing condition #2 (Table 3). The resulting microcapsules were stored at 4 °C under controlled humidity, as previously described, yet in this experiment storage time was duplicated to 60 days to assess extent of suitability of the method studied herein. Viability throughout storage and when subjected to in vitro simulated GIT conditions was assessed on the following timepoints: 1, 14, 28, 45, and 60 days. These results are discussed in the following section.

### 2.5. Viability Assessment of Microencapsulated A. Muciniphila upon Simulated GIT Conditions

In order to exert its beneficial effect as a probiotic in the host, a bacterial strain must reach the intestine in adequate viable numbers [11]. Thus, a good encapsulation system must be resistant to the harsh gastrointestinal conditions to deliver the probiotic strain in the optimal required conditions to the target location to yield the expected benefits. In the present study, the survivability of spray-dried *A. muciniphila* cells was assessed under simulated GIT conditions over a 60-day storage period on the following timepoints: 1, 14, 28, 45, and 60 days. As shown in Figure 4 (and Appendix A), spray-drying ensures more reliable results regarding the stabilization of viability over storage, as the time 0 h seems to indicate (Figure 4). Indeed, the viability decreased more abruptly from storage day 1 to day 14 and only slightly on the following intervals, which confirms the previously observed results.

Regarding the survivability of the microencapsulated *A. muciniphila* cells to the GIT conditions, the final cell counts remain mostly similar to those from the beginning of the GIT passage simulation (Figure 4 and Appendix A). A particular stabilization effect can be observed from storage from day 28 onwards, with the cell counts remaining steady throughout all the stages of the GIT passage (Figure 4 and Appendix A). Thus, these results clearly demonstrate that microencapsulation by spray-drying—using improved conditions and skim milk as a protective matrix—mitigates the effects of prolonged aerobic storage on *A. muciniphila* cell viability, while also providing resistance to the GIT passage. The study from Chang and colleagues also evaluated the tolerance of the spray-dried *A. muciniphila* to the gastric and intestinal fluids. These authors tested various matrices to improve the cell viability of *A. muciniphila* when exposed to the gastrointestinal fluids in comparison with the free cells; however, cell viability was assessed by optical density, which does not always correlate to the actual cell counts [20].

Until now, the exact number of live *A. muciniphila* cells required to exert a beneficial effect on humans was not determined. However, the oral administration of 10^8^ CFU of live *A. muciniphila* cells in murine models was sufficient to reverse high-fat diet-induced metabolic disorders [35]. In addition, an exploratory study performed by Depommier and colleagues demonstrated that the daily oral supplementation of 10^10^
*A. muciniphila* (either alive or pasteurized) for 3 months was not only safe and well-tolerated but also improved particular metabolic markers in overweight and obese human volunteers [10]. Although the recommended probiotic administration dosage is 10^9^ CFU/dose [11], probiotics effects have been demonstrated for administered dosages starting from 10^8^ CFU onwards [36]. Thus, it seems reasonable to assume that, in the present conditions, the survival of *A. muciniphila* can be ensured for up to 60 days, under refrigerated, aerobic storage—mimicking the more conventional shelf-life conditions—at levels that can meet the criteria recommended for a probiotic-containing product, a promising feature for its use as a live biotherapeutic.

## 3. Materials and Methods

### 3.1. Bacterial Strains and Growth Conditions

*Akkermansia muciniphila* DSM 22,959 strain was obtained from Leibniz Institute DSMZ-German Collection of Microorganisms and Cell Cultures (Braunschweig, Germany). For long-term storage, this bacterial strain was kept frozen at −80 °C in PYG broth supplemented with 0.05% (*m/v*) mucin (PYGM, media composition as recommended by DSMZ [37] except that no resazurin was added), with 20% (*v/v*) glycerol (Fisher Chemical, Loughborough, UK). For each experiment, a frozen vial of *A. muciniphila* DSM 22,959 was thawed and grown in PYGM broth for 24 h at 37 °C, under anaerobic conditions (85% N_2_, 5% H_2_ and 10% CO_2_—Whitley A35 HEPA anaerobic workstation, Bingley, UK). After incubation, the bacterial suspension was subcultured twice in PYGM broth to ensure high bacterial concentration. Cell biomass was harvested by centrifugation (Sorvall LYNX 4000, Thermo Scientific, MA, USA) at 12,000× *g*, for 30 min, at 4 °C and washed once with the same volume of sterile phosphate buffer saline (PBS × 1; VWR, Radnor, PA, USA). After centrifugation, the pelleted biomass was resuspended in 200 mL of physiological saline solution (NaCl; 0.9% *m/v*), to a final concentration of approximately 2 to 6 × 10^9^ colony forming units per milliliter (CFU/mL). The resulting bacterial suspension was either used directly for free cell survival determination or for spray-drying, as follows.

### 3.2. Spray-Drying of A. Muciniphila in Different Matrices and Drying Conditions

The encapsulating matrices used were: skim milk (SM; Oxoid, UK), whey protein concentrate (80%; WPC; Bulk Powders, UK), and whey protein isolate (90%; WPI; Bulk Powders, UK). All matrices were prepared at a final concentration of 10% (*m/v*), as follows: SM was dissolved in deionized water and sterilized by autoclaving at 121 °C for 5 min; WPC and WPI were prepared with previously autoclaved deionized water, no boiling required. The bacterial concentrate was added to each matrix at a final concentration of 10% (*v/v*). Two different temperatures were tested for both the inlet—150 and 170 °C—and outlet—65 and 75 °C.

A laboratory scale spray-dryer (B-290 Mini Spray-Dryer, Switzerland) was used, employing combinations of inlet and outlet air temperatures, considering previously determined conditions for other probiotic strains [21,22] (Table 3; the aspirator was set to 65% for all conditions). The sample was atomized into the drying chamber using a two-fluid nozzle and the product dried almost instantaneously, with a very low residence time. Spray-dried powders resulting from the different treatment combinations were collected and were mixed thoroughly with a sterile spatula. The dried powders were weighed and aerobically stored (in duplicates) using sealed 5 mL-centrifuge tubes, at two different temperatures: 4 ± 1 °C and 22 ± 1 °C. The storage humidity was maintained at around 33–34% using a saturated solution of magnesium chloride [26].

### 3.3. Water Activity

The water activity (a_w_) was measured using a HygroLab C1 (Rotronic, Bassersdorf, Germany) at room temperature (25 ± 1 °C). Spray-dried powders (approximately 1 g) were placed on the sample holder of the water activity measuring device; a sealed system was formed by placing the water activity probe on top of the sample holder. When a_w_ became constant (which usually took less than 1 h), the corresponding value was recorded. This assessment was performed immediately after production and after 28 days of aerobic storage under controlled humidity.

### 3.4. Viability Determination of Free and Spray-Dried A. Muciniphila Cells

For the determination of cultivable spray-dried bacteria, cells were previously released from the microparticles by disintegration using a pellet pestle mixer (Cordless^®^ Motor; Fisher Scientific, Hampton, NH, USA) for 30–60 s in PBS 1×. This treatment was previously tested and was shown not to affect the viability of *A. muciniphila* cells (unpublished results). From here, both cell suspensions (homogenized spray dried and non-encapsulated *A. muciniphila* cells) were treated equally: aliquots were serial diluted with PBS; 10 µL of each dilution were spotted, in triplicates, on PYGM agar plates (PYGM broth supplemented with 1.5% (*m/v*) agar (Liofilchem, Roseto degli Abruzzi, Italy)). The plates were incubated for 5–7 days at 37 °C under anaerobic conditions and results were expressed in CFU/mL, in the case of free cells for the initial suspensions controls, or CFU/g, for the spray dried cells.

*Akkermansia muciniphila* viability was also assessed, using the same methodology, throughout aerobic storage at 4 °C and 22 °C, after 7, 14, 21, and 28 days of storage.

### 3.5. Microcapsules Morphology

When adequate, microcapsules resulting from the spray-drying process were characterized in terms of morphology. Dried samples were mounted on double-sided adhesive carbon tabs mounted on Scanning Electron Microscopy (SEM) stubs and coated with gold/palladium and viewed under a JEOL 5600 (JEOL, Tokyo, Japan) scanning electron microscope at an accelerating voltage of 20 kV. Microcapsule size was determined using ImageJ (1.52 a; https://imagej.nih.gov/ij/), (accessed on 15 March 2022) resulting from the average of 400 measurements [38,39].

### 3.6. Viability Determination of A. Muciniphila Cells throughout Simulated GIT Conditions

The viability of free and spray-dried *A. muciniphila* in simulated GIT conditions was determined 1, 14, 28, 45, and 60 days after the drying process, using a standardized procedure described by [40], with some modifications. The cells, either in suspension or spray-dried, were maintained under aerobic refrigerated storage. Briefly, spray-dried *A. muciniphila* was weighed and rehydrated with 0.9% NaCl at room temperature as to produce a homogeneous solution, from which 0.5 mL were distributed into independent tubes (two replicates per timepoint per condition; weights and volumes of the powders and NaCl solution used, respectively, were later accounted for viability determination); 0.5 mL of free cells in 0.9% NaCl (*m/v*) were used as control. To replicate the temperature and peristaltic movements that prevail during human gastrointestinal transit, an orbital shaker incubator (Wiggen Hauser, Berlin, Germany) was used at 37 °C and 150 rpm. For each experiment, all enzyme solutions were freshly prepared. For the esophagus-stomach step (gastric phase), samples were exposed, for 2 h, to 2 mL of simulated gastric fluid (pH 3), containing pepsin (2000 U/mL—from porcine gastric mucosa; Sigma Aldrich, St. Louis, MO, USA), pH adjusted to 3 with 6 M HCl. After that, intestinal conditions were simulated for 3 h after the addition of 4 mL of simulated intestinal fluid containing pancreatin (based on the trypsin activity at 100 U/mL in the final mixture; Sigma Aldrich, St. Louis, MO, USA) and bile salts (Sigma Aldrich, St. Louis, MO, USA), with pH adjusted to 7 using either 6 M HCl or 4 M NaOH, depending on the starting pH of the mixture. Samples were collected every hour in order to follow the simulated gastric and intestinal effects: after 1 h and 2 h in the gastric phase and after 1, 2, and 3 h in the intestinal phase. For each sampling point, cell viability was determined in CFU/mL or CFU/g, according to procedure previously described. The protocol was performed under aerobic conditions, whilst the incubation of PYGM plates was performed under anaerobic conditions.

### 3.7. Statistical Analyses of Experimental Data

Data were expressed as mean of CFU/mL or CFU/g ± standard deviation (SD). One- or two-way analysis of variance (ANOVA) tests were performed with SigmaStat™ 3.1 (Systat Software, Chicago, IL, USA), with results considered significant when *p* < 0.05. Normality and homoscedasticity of data were validated. Whenever normality of data was not verified, a Kruskal–Wallis one-way ANOVA on Ranks was applied at the same level of significance. Holm-Sidak method or Tukey test were used for pairwise comparisons.

## 4. Conclusions

The applicability of spray-drying technology to the live biotherapeutic candidate *A. muciniphila* was evaluated. Different factors were analyzed, namely three protective matrices—skim milk, whey protein concentrate, and whey protein isolate—and the combination of inlet and outlet operating temperatures. The resulting formulations were immediately assessed in terms of production yield, survivability, and water activity. Additionally, these formulations were stored under aerobic conditions and controlled humidity, at two common storage temperatures—4 and 22 °C—and *A. muciniphila* viability was assessed over time. Considering all the tested conditions, the best results regarding encapsulation efficiency and viability after processing were obtained when *A. muciniphila* was spray-dried using a suspension of 10% skim milk and a combination of inlet/outlet temperatures of 150/65 °C. In addition, this matrix allowed the maintenance of *A. muciniphila* viability up to 28 days, stored at 4 °C, under aerobic conditions and controlled humidity, with viability losses never reaching 1 log reduction. Finally, this formulation also provided resistance to *A. muciniphila* under simulated GIT conditions, particularly after prolonged storage periods. Although further optimization is needed, in particular for scale-up purposes, these results show that spray-drying technology, using easily available and economical matrices, can be successfully applied to produce microcapsules containing live *A. muciniphila* DSM 22,959 with high enough yields to encourage a future investment in this field of research. Moreover, contrarily to other formulations developed so far, it provides stability to the potential probiotic formulation under standard storage conditions. Ultimately, this methodology enables the putative delivery of the recommended dosage of live bacteria in the human gut, where it is expected to confer a health benefit to the host.

## Figures and Tables

**Figure 1 pharmaceuticals-15-00628-f001:**
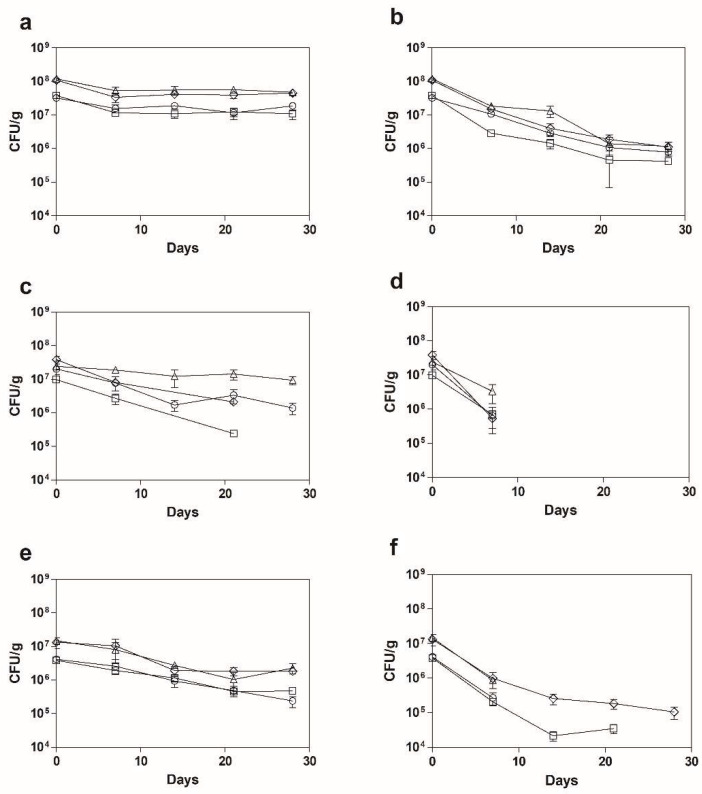
Viability of spray dried *A. muciniphila* cells throughout aerobic storage. Cells were processed in SM (**a**,**b**), WPC (**c**,**d**) and WPI (**e**,**f**), using the previously described four conditions: #1—circles; #2—triangles; #3—diamonds; #4—squares. Samples were stored for 28 days at 4 (**a**,**c**,**e**) or 22 °C (**b**,**d**,**f**), under aerobic atmosphere and controlled humidity.

**Figure 2 pharmaceuticals-15-00628-f002:**
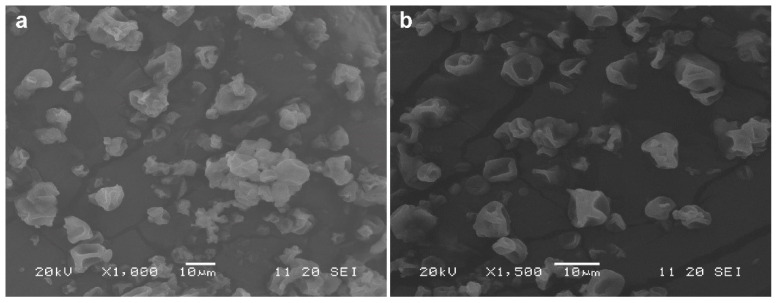
SEM micrographs of spray-dried *A. muciniphila*, encapsulated within a matrix containing 10% skim milk, using selected drying conditions (inlet/outlet temperatures: 150/65 °C); (**a**) scale bar 10 µm, magnification 1000×; (**b**) scale bar 10 µm, magnification 1500×.

**Figure 3 pharmaceuticals-15-00628-f003:**
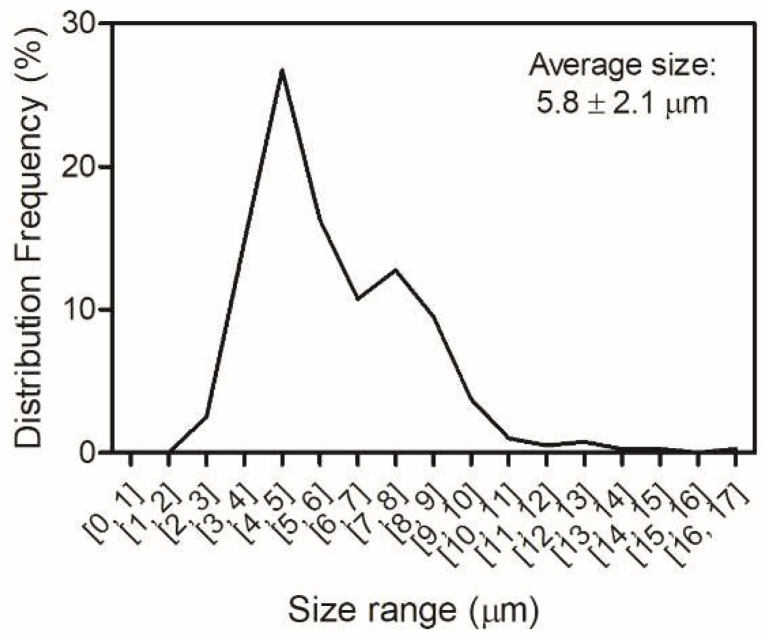
*Akkermansia muciniphila* DSM 22,959 microcapsules size relative distribution (µm). The average size is also presented. These values result from 400 independent measurements within the sample. Matrix: 10% skim milk; inlet/outlet temperatures: 150/65 °C.

**Figure 4 pharmaceuticals-15-00628-f004:**
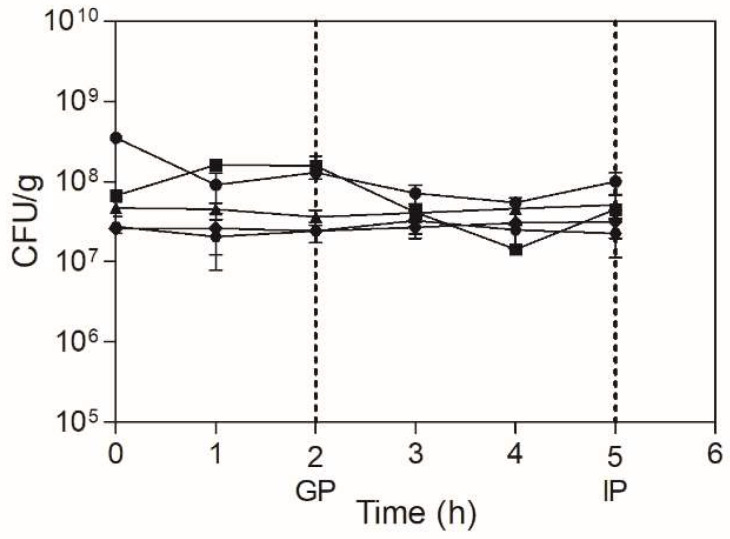
Viability of spray-dried *A. muciniphila* DSM 22,959 exposed to simulated GIT conditions for 5 h. The assessment was performed at defined timepoints after aerobic, refrigerated storage: 1 (circles); 14 (squares); 28 (triangles); 45 (hexagons); and 60 (diamonds) days. GP = end of the gastric phase; IP = end of the intestinal phase.

**Table 1 pharmaceuticals-15-00628-t001:** Final masses, in grams, and *A. muciniphila* viable cells obtained for each protective matrix and condition tested. Total number of cells incorporated (“Initial”) into each matrix and recovered in each condition. CFU = colony-forming units; SD = standard deviation. For further information regarding the settings of conditions #1 to #4, check Table 3 in Section 3.2 of Material and Methods.

Conditions ^1^	SM (10%)			WPC (10%)			WPI (10%)		
	Recovery Yield (%)	Total CFU ± SD	CFU/g	Recovery Yield (%)	Total CFU ± SD	CFU/g	Recovery Yield (%)	Total CFU ± SD	CFU/g
Initial	-	1.35 ± 0.25 × 10^10^	-	-	1.16 ± 0.11 × 10^11^	-	-	5.57 ± 0.50 × 10^10^	-
#1	66.5	2.15 ± 0.23 × 10^8^	3.23 ± 0.34 × 10^7^	69.0	1.38 ± 0.21 × 10^8^	2.00 ± 0.30 × 10^7^	74.4	3.05 ± 0.53 × 10^7^	4.10 ± 0.71 × 10^6^
#2	68.1	8.17 ± 1.47 × 10^8^	1.20 ± 0.22 × 10^8^	65.9	1.58 ± 0.67 × 10^8^	2.40 ± 1.02 × 10^7^	79.9	1.17 ± 0.26 × 10^8^	1.47 ± 0.33 × 10^7^
#3	61.1	6.72 ± 1.22 × 10^8^	1.10 ± 0.20 × 10^8^	71.5	2.74 ± 0.74 × 10^8^	3.83 ± 1.03 × 10^7^	77.7	1.04 ± 0.37 × 10^8^	1.33 ± 0.47 × 10^7^
#4	49.4	1.90 ± 0.16 × 10^8^	3.83 ± 0.33 × 10^7^	65.8	6.38 ± 0.10 × 10^7^	9.70 ± 0.01 × 10^6^	77.4	2.94 ± 0.46 × 10^7^	3.80 ± 0.59 × 10^6^

^1^ Inlet/Outlet temperatures: #1 = 150/75 °C; #2 = 150/65 °C; #3 = 170/65 °C; #4 = 170/75 °C.

**Table 2 pharmaceuticals-15-00628-t002:** Water activity measured for each protective matrix and condition tested, immediately after processing and upon 28 days of storage at 4 and 22 °C, under controlled humidity. The values correspond to the average of two independent replicates ± standard deviation. For further information regarding the settings of conditions #1 to #4, check Table 3 in Section 3.2 of Material and Methods.

Conditions ^1^	SM (10%)			WPC (10%)			WPI (10%)		
	Day 0	Day 28		Day 0	Day 28		Day 0	Day 28	
		4 °C	22 °C		4 °C	22 °C		4 °C	22 °C
#1	0.261 ± 0.003	0.272 ± 0.025	0.236 ± 0.018	0.234 ± 0.015	0.277 ± 0.016	0.323 ± 0.035	0.268 ± 0.005	0.309 ± 0.018	0.330 ± 0.015
#2	0.286 ± 0.003	0.304 ± 0.008	0.279 ± 0.000	0.276 ± 0.009	0.310 ± 0.004	0.338 ± 0.001	0.330 ± 0.006	0.384 ± 0.024	0.355 ± 0.008
#3	0.246 ± 0.002	0.265 ± 0.002	0.260 ± 0.018	0.307 ± 0.007	0.335 ± 0.017	0.346 ± 0.003	0.331 ± 0.007	0.360 ± 0.005	0.355 ± 0.009
#4	0.257 ± 0.005	0.308 ± 0.005	0.265 ± 0.012	0.261 ± 0.010	0.306 ± 0.002	0.333 ± 0.013	0.296 ± 0.001	0.362 ± 0.011	0.340 ± 0.010

^1^ Inlet/Outlet temperatures: #1 = 150/75 °C; #2 = 150/65 °C; #3 = 170/65 °C; #4 = 170/75 °C.

**Table 3 pharmaceuticals-15-00628-t003:** Spray-drying conditions used in this study. Based on [21,22].

Conditions	Inlet (°C)	Outlet (°C)	Pump Speed (%)	Flow Rate (mL/min)
**SM (10%)**				
#1	150	75–77	20	6
#2	150	65	35	10.5
#3	170	65–66	45	13.5
#4	170	75	30	9
**WPC (10%)**				
#1	150	75	20	6
#2	150	65	25	7.5
#3	170	65–67	35	10.5
#4	170	75–76	25	7.5
**WPI (10%)**				
#1	150	75–76	20	6
#2	150	66	30	9
#3	170	65–67	35	10.5
#4	170	74–75	25	7.5

## Data Availability

Data is contained within the article and Appendix A.

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
