# Peer review of "Spray-Drying Encapsulation of the Live Biotherapeutic Candidate *Akkermansia muciniphila* DSM 22959 to Survive Aerobic Storage"

_pharmaceuticals, 2022, doi:10.3390/ph15050628_

Round 1

Reviewer 1 Report

This manuscript describes spray-drying encapsulation of the live probiotic Akkermansia muciniphila. My specific comments are as follows:

(1) This study does not include in vivo evaluation study. In vivo efficacy study needs to be performed for demonstrating the viability of the formulations. 

(2) The authors measured the size of microcapsules by ImageJ. In general, the laser diffraction method is recommended for the size measurement of microparticles. 

(3) Table 3 should be appropriately positioned following the title in page 9.

Author Response

Comment (C)1: This study does not include in vivo evaluation study. In vivo efficacy study needs to be performed for demonstrating the viability of formulations.

Answer (A)1: The authors thank the reviewer for the important comment and acknowledge that in vivo studies are indeed important to understand the behavior of probiotic strains upon formulation and following uptake. However, this is a necessary preliminary work regarding the selection of the most favorable spray-drying parameters to ensure the viability in standard conditions. In vivo studies using a suitable model will be required in order to confirm viability and, more importantly the therapeutical efficacy, and thus, are out of the scope of this study

C2: The authors measure the size of microcapsules by ImageJ. In general, the laser diffraction method is recommended for the size measurement of microparticles.

A2: The authors thank the reviewer for the pertinent comment and acknowledge the recommended use of laser diffraction to measure the size of the particles; however, in the present work, it is intended to have an indication of the particle size distribution rather than a more precise measure. As so, it was decided to take advantage of the images acquired with SEM and use them to estimate the particle size and respective distribution. References of previous works that use the same approach were included in the manuscript (doi:10.3390/FOODS11081111 and doi:10.3390/PHARMACEUTICS12060496).

C3: Table 3 should be appropriately positioned following the title in page 9.

A3: The authors thank Reviewer 1 for this comment. It was changed accordingly.

Reviewer 2 Report

In the study, spray drying was applied for encapsulation of probiotic A. muciniphila. Three different carriers were used, and drying conditions were changed. Study provides significant insights, and I agree that this topic deserves to be investigated. However, I have some recommendations, as follows:

Abstract is very general and my recommendation it to rewrite it. Apart from the goal of the study, please include methods and material, results, and main findings. Also, include quantitative results.

Instead of final mass, my recommendation is to include the recovery rate/yield of drying in %.

Please include an explanation of how conditions were chosen, based on literature or preliminary experiments.

It is concluded that, quote: “higher protective effect during processing, probably since both inlet and outlet temperatures are milder”. Is it possible to test even lower temperatures?

How to decrease water activity and provide safer long-time storage?

Also, try to elaborate more on the influence of temperature and other drying conditions.

Author Response

C1: Abstract is very general and my recommendation is to rewrite it. Apart from the goal of the study, please include methods and material, results and main findings. Also, include quantitative results.

A1: The authors are thankful for the reviewer´s suggestion which has enabled us to objectively improve our abstract. Abstract was rewritten and changes are marked with the “Track Changes” function.

C2: Instead of final mass, my recommendation is to include the recovery rate/yield of drying in %.

A2: The authors acknowledge and thank the reviewer´s recommendation, and it was changed accordingly.

C3: Please include an explanation of how conditions were chosen, based on literature or preliminary experiments.

A3: The authors thank the reviewer´s suggestion. The references from where these conditions were based on were added in the Material and Methods and Results and Discussion sections.

C4: It is concluded that, quote: “higher protective effect during processing, probably since both inlet and outlet temperatures are milder”. Is it possible to test even lower temperatures?

A4: The principle of spray drying relies on finding the ideal temperature to ensure the rapid drying of the sample while inducing the least possible bacterial damage by heat. Decreasing the temperature will lead to higher water activity and increased moisture in the samples, which in turn will possibly decrease the viability over time and reduce the shelf-life of the formulation. A brief explanation was included within the manuscript regarding this matter. Thus, lower temperatures could render the process non-effective, not allowing a successful drying of the matrix.

C5: How to decrease water activity and provide safer long-time storage?

A5: In line with the previous comment, one possible way to reduce water activity would be to increase the processing temperatures; however, that would potentially induce higher heat damage in the bacterial cells. The evaluation of methods other than the optimization of the drying parameters is out of the scope of the present work.

C6: Also, try to elaborate more on the influence of temperature and other drying conditions.

A6: The authors thank the reviewer for the recommendation, and additional discussion regarding these parameters was added, with reference to a review paper.

Reviewer 3 Report

Dear authors,
The article presented for review raises important issues related to immunity related to our intestinal microflora. I find the selected strain of bacteria interesting. The research presented in this article is valuable and promising. The justification of the research results is correct and clear.
The only caveat / question I have regarding the spray drying methodology - why were different flow rates used?
Please also pay attention to the position of table 3 in the text of the article.

Author Response

C1: The only caveat/question I have regarding the spray drying methodology – why were different flow rates used?

A1: The authors thank the reviewer for the question which allows us to clarify better our work. In the present work, the authors intended to work at defined inlet and outlet temperatures. The spray drying set up allows the setting of the inlet temperature and the flow rate, but not the outlet temperature, which is a direct consequence of the combination of both the inlet temperature and the flow rate. Thus, to adjust the outlet temperature for the one required, different flow rates must be set depending on the inlet temperature.

C2: Please also pay attention to the position of Table 3 in the text of the article.

A2: The authors thank Reviewer 3 for this comment. It was changed accordingly.

Reviewer 4 Report

The manuscript entitled “Spray-drying encapsulation of the live biotherapeutic candi- 2 date Akkermansia muciniphila DSM 22959 to survive aerobic 3 storage” describes the methodology to preserve Akkermansia muciniphila DSM 22959 against storage and GIT conditions. It’s a very interesting work in developing Akkermansia muciniphila DSM 22959 as biotherapeutic. Here are a few comments that can improve the manuscript.

  • There no statistical analysis is reported in the manuscript especially when the author determining the conditions that gave optimal CFU/g (Table 1) and the survival of the cells under aerobic storage conditions (Figure 1).
  • Table 1 is quite confusing since #1, #2, #3 and #4 are not explained nor in table or legend.
  • “The main strength of the present study is that muciniphila viability was maintained at levels rounding 107 CFU/g after 28 days of refrigerated storage, similarly to that reported by previous authors, yet such occurred under aerobic conditions, in contrast to the anaerobic conditions used by other authors (of higher cost)” should have citation since the author comparing with previous studies.

Overall I believe this manuscript is suitable for publication after minor modification.

Author Response

 C1: There no statistical analysis is reported in the manuscript especially when the author is determining the conditions that gave optimal CFU/g (Table 1) and the survival of the cells under aerobic storage conditions (Figure 1).

A1: The authors thank Reviewer 4 for this comment. Statistical analysis was indeed missing and was added to the manuscript where appropriated.

C2: Table 1 in quite confusing since #1, #2, #3 and #4 are not explained nor in table or legend.

A2: The authors are thankful to the reviewer for the pertinent comment. The conditions are defined in the section of Material and Methods. However, because Table 1 should effectively stand on its own, this information was added in the Table caption.

C3: “The main strength of the present study is that municiphila viability was maintained at levels rounding 107 CFU/g after 28 days of refrigerated storage, similarly to that reported by previous authors, yet such occurred under aerobic conditions, in contrast to the anaerobic conditions used by other authors (of higher cost)” should have citation since the author comparing with previous studies.

A3: The authors thank the reviewer´s suggestion. This sentence comes as a wrapping up related with the comparisons made in the paragraph immediately before, where all the previous works are cited, described and compared with the results of the present work. This has now been made clearer by changing the beginning of the sentence and by adding it to the previous paragraph. The whole paragraph itself was re-written to improve clarity.

Round 2

Reviewer 2 Report

The authors revised and corrected the Manuscript according to the suggestions. Therefore, I recommend accepting of the manuscript.